# Maladaptive Cognitive Schemas as Predictors of Disordered Eating: Examining the Indirect Pathway through Emotion Regulation Difficulties

**DOI:** 10.3390/ijerph191811620

**Published:** 2022-09-15

**Authors:** Sarah Gerges, Souheil Hallit, Diana Malaeb, Sahar Obeid

**Affiliations:** 1School of Medicine and Medical Sciences, Holy Spirit University of Kaslik, Jounieh P.O. Box 446, Lebanon; 2Psychology Department, College of Humanities, Effat University, Jeddah 21478, Saudi Arabia; 3Research Department, Psychiatric Hospital of the Cross, Jal Eddib P.O. Box 60096, Lebanon; 4College of Pharmacy, Gulf Medical University, Ajman 20550, United Arab Emirates; 5Social and Education Sciences Department, School of Arts and Sciences, Lebanese American University, Jbeil 1401, Lebanon

**Keywords:** eating attitudes, inappropriate eating, dysfunctional cognitions, early maladaptive schemas, maladaptive core beliefs, emotion regulation, emotion dysregulation, young adults

## Abstract

A scarcity of research has looked into the association of maladaptive core beliefs with dysfunctional eating patterns. Moreover, no prior study has considered the potential role of difficulties in negative emotion regulation when disentangling the complex correlations between early maladaptive schemas and disturbed eating habits. Our study aimed at exploring the distinct relationships between early maladaptive schemas and disordered eating, while investigating the indirect role of emotion regulation difficulties within these associations. We collected data from 982 Lebanese young adults (18–30 years old), distributed across the five Lebanese governorates, who completed the Eating Attitudes Test (EAT-26), the Young Schema Questionnaire—Short Form 3 (YSQ-SF3), and the Difficulties in Emotion Regulation Scale—16 Item Version (DERS-16). The results showed that the disconnection and rejection schema domain, under which the early maladaptive schema of mistrust was the most predictive of disordered/inappropriate eating attitudes. All the remaining maladaptive schema domains (i.e., the impaired autonomy/performance, impaired limits, other-directedness, and overvigilance/inhibition schema domains) exerted significant indirect effects on disordered eating attitudes through difficulties in emotion regulation. Our findings gave prominence to a potential intrinsic mechanism through which maladaptive cognitive schemas are linked to disordered eating behaviors, emphasizing the role of emotion dysregulation as a cardinal actor within this model. They sustain the surmise that cognitively and emotionally vulnerable individuals exhibit stronger propensities for inappropriate dietary patterns, as a means to offset their inner weakness. This study broadens the medical community’s insights into the underpinning processes behind eating disorder psychopathology and could therefore make a step towards the adoption of innovative therapeutic approaches that promote emotion regulation skills in the context of schema therapy.

## 1. Introduction

During the past decades, eating disorders have been on the rise in our ever-changing environment, affecting people from all social backgrounds and geographies [1,2]. These conditions have been constantly acknowledged as impairing, even lethal, mental disorders, which alter physical functions and carry substantial psychosocial costs [3] as a result of dysfunctional attitudes towards body weight, size and shape, and eating patterns [2]. Although individuals from all ages might be prone to such disorders, adolescents and young adults constitute a vulnerable group [2,4]. Nonetheless, eating disorders remain a broadly understudied field, and high levels of uncertainty around their physiopathology have been pointed out [2]. In addition, notwithstanding a global upsurge in prevalence, only a minority of impacted individuals would seek therapeutic interventions [5,6]. Thus, integrating eating habits evaluation in routine health assessment seems judicious [2], and advances in eating disorder research are still warranted to effectively portray the core behavioral psychopathological models.

Previously conducted research in multiple nations and cultural settings, including the Lebanese population, has emphasized the correlations between eating pathology and psychological distress, body dissatisfaction, personality traits/disorders, and affect-related difficulties [7,8,9,10,11]. However, when studying the psychopathology of disordered eating development, further overriding factors to explore are dysfunctional core beliefs [12]. Indeed, the literature has accentuated that disorder-specific maladaptive cognitive profiles (i.e., ruminative thinking and excessive worry towards the inability to control weight, body shape, and eating styles) [13,14,15], together with underpinning dysfunctional cognitive–affective processes, are sharply incriminated in the induction and perpetuation of pathological eating [16].

Schemas (or “schemata”) were first introduced by Beck [13,17], who defined them as disturbed thinking patterns, contingent on prior life events, whose activation conditions insights and interpretation of the surrounding environment. In 1982, Garner and Bemis postulated that in eating disorders, individuals witness the activation of uncontrollable negative thoughts, such as the vitalness of slimness and the need for eating avoidance, subsequent to disordered underlying assumptions about the enthralling nature of thinness and the uncontrollability of danger when gaining weight [18]. Nevertheless, several theorists have conjectured that underlying disorder-specific cognitions alone would fail to completely uncover the cognitive bases of psychopathological eating [19,20,21]. Yet, at that time, another interesting theory to explore emerged—a perspicacious expatiation of Beck’s conceptualization of core beliefs—the Young Schema theory, which introduced the construct of early maladaptive schemas, built on clinical analyses of personality disorders [22].

Early maladaptive schemas are conceptualized as dysfunctional and pervasive themes, established from early childhood/adolescence as a consequence of negative experiences with significant others, including core beliefs, emotions, physical sensations, and memories that unconsciously dictate individuals’ future models of thinking and acting in regard to themselves and their relations with others and the interacting environment [22,23]. Their triggering, however, is maintained throughout life through various occurrences unconsciously perceived as analogous to childhood trauma. Unlike dysfunctional cognitive representations about eating and weight (e.g., “I am a failure because I am fat”), maladaptive core beliefs are broader models of disordered thinking, representing rigid and unconditional negative beliefs about the self (e.g., “I am a failure”) [12]. Thus, as speculated by Young, when activated, early maladaptive schemas concretize the most culminant and profound dimension of cognition and affect [23]. In eating disorders, such maladaptive core beliefs are hypothesized to be one of the core essences of developing disturbed thoughts regarding shape and weight, in an attempt of schema compensation [22]. In fact, the theory suggests that gaining control over these facets of eating would help individuals conquer pejorative self-beliefs and thus compensate for their schema activation [12,24].

Accordingly, several studies have revealed more pronounced scores of early maladaptive schemas among people with symptoms of eating disorders [25,26]. For instance, Unoka et al. observed substantial relationships between eating disorder behaviors and early maladaptive schemas, thus supporting the hypothesis that dimensions of maladaptive cognitive schemas play a significant role in the development and maintenance of eating behavior disturbances [27]. Likewise, Elmquist et al. witnessed increased overall levels of early maladaptive schemas in patients with a probable eating disorder compared to controls [28]. Consistently, a multitude of researchers found that in contrast to healthy controls, patients with anorexia nervosa manifested a significant potentiation of maladaptive schemas [29,30]. Nonetheless, despite the available data spotlighting direct links between early maladaptive schemas and numerous eating disorders symptoms, a recent synthesis of existing research has pointed out the shortfall of insights into the psychological mechanisms governing these associations, hence underscoring the paramount importance of testing mediating variables as part of a future research direction [26].

From this perspective arises another riveting multifaceted construct to consider for a new speculative composite equation—difficulties in regulating negative emotional states. Such struggles are immensely correlated with psychopathologies [31]; thus, in concordance with the principal goal of schema therapy [32], gaining emotion regulation assets would be the initial step towards an adaptive treatment strategy [33]. In fact, while emotion regulation is commonly understood as a cognitive and behavioral resilience to the loss of self-control when facing intensely disagreeable emotions, entailing adequate management of adverse circumstances, this concept additionally encompasses emotional awareness, comprehension, and acceptance, transcending the sole process of attenuating emotions [34]. The literature has previously emphasized the involvement of emotion dysregulation in instigating and maintaining eating disorders [35]. To exemplify, inferior emotional clarity and awareness were reported among eating disorder patients; moreover, researchers highlighted that emotional unconsciousness, emotional refusal, and deficient practical emotion regulation strategies could all predict cognitive eating disorder misrepresentations and symptoms. Therefore, emotion dysregulation has been overall labelled as a pivotal “transdiagnostic” feature of eating disorders [35,36].

Further, as for the association between early maladaptive schemas and emotional regulation, according to Young et al., the prominence of early maladaptive schemas enhances emotion dysregulation [23]. In addition, interestingly, although schema therapy does not straightly target emotion regulation, its treatment techniques imply systematic interventions that indirectly address the regulation of negative affect [32,37]. Namely, early maladaptive schemas would dictate low emotional coping responses, triggering a plethora of powerful negative emotions, hence leading individuals to psychopathological disorders [31]. Nevertheless, in regard to eating disorders in particular, to date, no empirical analysis has assessed this conjecture.

Thus far, a scarcity of research has looked into the association of maladaptive core beliefs with dysfunctional eating patterns. Moreover, to our best knowledge, no prior study has considered the potential role of difficulties in negative emotion regulation when disentangling the complex correlations between early maladaptive schemas and disturbed eating habits. Therefore, our study aimed at exploring the distinct relationships between early maladaptive schemas and disordered eating (i.e., tendencies for bulimia and anorexia symptoms) in a sample of Lebanese young adults, while investigating the indirect role of emotion regulation difficulties within these associations. We worked towards identifying the maladaptive schemas/schema domains that would best predict eating behavior disorders; additionally, we hypothesized that emotion regulation problems act as a key mediating factor between such dysfunctional cognitive aspects and eating pathology.

## 2. Materials and Methods

### 2.1. Study Design and Participants

This observational study took place in a three-month period, from September to December 2020. We collected data from 982 Lebanese young adults, aged between 18 and 30 years, distributed across the five Lebanese governorates (i.e., Beirut, Mount Lebanon, North Lebanon, South Lebanon, and Bekaa). The online snowball sampling technique was used to recruit the respondents, as this study was conducted during the COVID-19 lockdown. They could access the questionnaire via a Google Form link shared on social media networks (i.e., WhatsApp, Facebook, and LinkedIn applications). All gathered responses were anonymous.

A total of 982 participants enrolled in our study, with a mean age of 21.97 ± 3.33 years and 701 (71.4%) females. All characteristics of the sample are displayed in Table 1.

### 2.2. Minimal Sample Size Calculation

Schönbrodt and Perugini have demonstrated that for estimating correlations in psychological research, stability is achieved when the sample size approaches a number of 250 participants [38]. Ultimately, 982 young adults were engaged in our study.

### 2.3. Translation Procedure

As the Arabic version of the Eating Attitudes Test (EAT-26) was previously validated in Lebanon [8], the translation procedure concerned the remaining self-report measures. This process was executed via two steps: a forward-translation into Arabic, the native language of Lebanon, and a back-translation into English. These two translations were performed by two distinct clinical psychologists. The initial English scales and the back-translated versions were analyzed to detect dissimilarities and fix inconsistencies. An expert committee constituted of psychologists and psychiatrists assented to the final Arabic versions of the scales.

### 2.4. Questionnaire and Variables

The estimated time to complete the questionnaire was 20 min. The first part gathered data about sociodemographic characteristics, such as age, educational level (i.e., secondary or less, or university), marital status, governorate of residence, and economic status. The latter was quantified by the household crowding index, which was calculated by dividing the number of people (including the respondent) by the number of rooms in the respondent’s house (other than the bathrooms and kitchen) [39]. The physical activity index was computed by multiplying the frequency, intensity, and duration of exercising [40].

The following measures were included in the second part:**1.** **The Eating Attitudes Test (EAT-26)**

The EAT-26 is a 6-point Likert tool that detects disordered eating behaviors (labeled as “inappropriate” eating) as well as preoccupations about weight. Its 26 questions scrutinize patterns of dieting (e.g., “Avoid eating when I am hungry”), bulimic tendencies, and preoccupation about food (e.g., “Am preoccupied with the thought of having fat on my body”), as well as oral control attitudes (e.g., “I have the impulse to vomit after meals”). Response options range from “never” to “always”. Individuals who score higher manifest stronger disordered eating symptoms [41,42]. This scale is validated in Lebanon [8].

**2.** 
**The Young Schema Questionnaire—Short Form 3 (YSQ—SF3)**


Within this study, we used the third version of the Young Schema Questionnaire [43]. This 90-item tool tests for the eighteen most common maladaptive schemas. Schemas were grouped into domains by Young and Brown [44]. Each schema is captured by five items, and each item’s response ranges from “completely untrue for me” to “describes me perfectly”. Scores reflect the intensity of schema activation. Following are the different schema domains and their respective early maladaptive schemas, presented along with sample items:

**A. The Disconnection and Rejection Schema Domain:** 1: Abandonment/Instability schema (e.g., “I find myself clinging to people I’m close to because I’m afraid they’ll leave me”); 2: Mistrust/Abuse schema (e.g., “I feel that people will take advantage of me”); 3: Emotional Deprivation schema (e.g., “I haven’t had someone to nurture me, share him/herself with me, or care deeply about everything that happens to me”); 4: Defectiveness/Shame schema (e.g., “No man/woman I desire could love me once he/she saw my defects”); and 5: Social Isolation schema (e.g., “I don’t fit in”).

**B. The Impaired Autonomy/Performance Schema Domain:** 1: Dependence schema (e.g., “I do not feel capable of getting by on my own in everyday life”); 2: Vulnerability to Harm schema (e.g., “I can’t seem to escape the feeling that something bad is about to happen”); 3: Enmeshment schema (e.g., “My parents and I tend to be over involved in each other problems”); and 4: Failure schema (e.g., “I’m incompetent when it comes to achievement”).

**C. The Impaired Limits Schema Domain:** 1: Entitlement/Grandiosity schema (e.g., “I am special and shouldn’t have to accept many of the restrictions placed on other people”); and 2: Insufficient Self-Control schema (e.g., “I can’t force myself to do things I don’t enjoy, even when it is for my own good”).

**D. The Other-Directedness Schema Domain:** 1: Subjugation schema (e.g., “I’ve always let others make choice for me, so I really don’t know what I want for myself”); 2: Self-Sacrifice schema (e.g., “I’ve always been the one who listens to everyone else’s problems”); and 3: Approval Seeking schema (e.g., “Accomplishments are most valuable to me if other people notice them”).

**E. The Overvigilance/Inhibition Schema Domain:** 1: Negativity/Pessimism schema (e.g., “Even when things seem to be going well, I feel that it is only temporary”); 2: Emotional Inhibition schema (e.g., “I control myself so much that people think I am unemotional”); 3: Unrelenting Standards schema (e.g., “I try to do my best; I can’t settle for “good enough”); and 4: Punitiveness schema (e.g., “I’m a bad person who deserves to be punished”).

**3.** 
**The Difficulties in Emotion Regulation Scale—16 Item Version (DERS-16)**


The Difficulties in Emotion Regulation Scale (DERS), which examines emotion regulation difficulties among adults, was first developed by Gratz and Roemer in 2004 [34]. Thereafter, the brief version (DERS-16) proved its reliability for effectively assessing emotion dysregulation [45]. The assessment of emotion regulation difficulties is performed by evaluating emotional clarity (via two items; e.g., “I am confused about how I feel”), the aptitude to confront emotional upsets and adopt goal-dictated behaviors (via three items; e.g., “When I am upset, I have difficulty thinking about anything else”), the virtue of controlling impulses (via three items; e.g., “When I am upset, I feel out of control”), the potential of applying effective emotion regulation strategies (via five items; e.g., “When I am upset, I believe that there is nothing I can do to make myself feel better”), and the capacity of accepting one’s emotional responses (via three items; e.g., “When I am upset, I feel like I am weak”). This scale has five response options, ranging from “almost never” to “almost always”. Higher scores indicate greater difficulties in regulating one’s emotions [45].

### 2.5. Statistical Analysis

Data analysis was conducted using the SPSS software version 23. As the option “required” was set for all questions on Google Forms, we did not encounter missing data. Cronbach’s alpha was computed for all scales and subscales. The normality of distribution of the age, physical activity index, maladaptive schemas/schema domains, and eating attitudes test scores were confirmed via a calculation of the skewness and kurtosis (between −2 and +2) [46]. Since the household crowding index was not normally distributed, it was transformed into log10 value; the latter showed normal distribution and was therefore used in the rest of the analysis. Weighting was performed for the general population according to gender and education level. The Student’s *t*-test was used to compare two means. The Pearson correlation test was used to correlate the EAT-26 score and continuous variables. In psychological research, a small effect size corresponds to correlations of 0.1, whereas medium and large effect sizes correspond to correlations of 0.2 and 0.3, respectively [47]. A linear regression was conducted, taking the EAT-26 score as the dependent variable; independent variables entered in the model were those that showed a *p* < 0.25 in the bivariate analysis.

The PROCESS SPSS Macro version 3.4 model four [48] computed the indirect effect of each maladaptive schema domain (as assessed by the YSQ-SF3) on disordered eating attitudes (EAT-26 scores) through difficulties in negative emotion regulation (DERS-16 scores), knowing that significant indirect effects are conditioned by bias-corrected bootstrapped 95% confidence intervals (CI) that are devoid of zero around them [48]. Statistical significance was deemed achieved for a two-sided *p* < 0.05.

## 3. Results

### 3.1. Reliability Analysis of the Scores

All scores showed excellent Cronbach’s alpha values (Table 2).

### 3.2. Bivariate Analyses

Bivariate analyses results are summarized in Table 3 and Table 4. All schemas were significantly associated with higher EAT-26 scores (more disordered/inappropriate eating), whereas a higher household crowding index (r = −0.13) was significantly associated with lower EAT-26 scores (more appropriate eating).

### 3.3. Multivariable Analyses

The early maladaptive schema of mistrust (beta = 0.55) (Table 5, Model 1) and the disconnection and rejection schema domain (beta = 0.11) (Table 5, Model 2) were significantly associated with higher EAT-26 scores (more inappropriate eating).

### 3.4. Indirect Pathways: Mediation Analysis

The Impaired Autonomy/Performance, Impaired Limits, Other-Directedness, and Overvigilance/Inhibition schema domains exerted significant indirect effects on disordered eating attitudes through difficulties in emotion regulation (Table 6; Figure 1, Figure 2, Figure 3 and Figure 4).

## 4. Discussion

Our study’s objective was to provide a deeper understanding of the cognitive and psychological predictors of disordered eating attitudes—an important yet unplumbed question. Ultimately, the analysis has enlightened a new psychopathological equation, involving cognitive distortions, emotional dysregulation, and disordered eating behaviors.

Indeed, the results showed that the disconnection and rejection schema domain was directly associated with an increased propensity for disordered eating attitudes. Cognitions regarding disconnection and rejection illustrate problematic attachment-related representations [49] (i.e., attachment instability). This schema domain characterizes individuals who are unable to securely establish satisfactory connections with their peers and entourage, bearing the conviction that their vital interhuman needs, such as respect, acceptance, integration, attention, love, emotional sharing, and protection, will never be contented [23]. Our finding fits into the broader literature that sheds light on the relation of insecure attachment with eating disorder symptoms; those disorders would serve as maladaptive coping processes against negative self-perceptions, thus helping individuals achieve a “perfect” self-image [50,51]. More specifically, a number of studies have established specific relationships between early maladaptive schemas of disconnection and rejection (e.g., emotional deprivation, abandonment, and social isolation) and disordered eating behaviors [27,29].

Precisely, the mistrust/abuse schema, under the disconnection and rejection domain, turned out to be most characteristic of young adult disordered eating in our sample, suggesting that this schema pattern might constitute a prominent mechanism in eating disorders psychopathology. Mistrust designates the apprehension of other’s intentions; namely, it embodies a perpetual expectation of cheat, imposture, abuse, humiliation, manipulation, or harm emanating from others in an intentional manner or as a result of excessive negligence [23]. Remarkably, patients with a diagnosis of anorexia nervosa have been recognized to display mental and psychological profiles that endorse self and interpersonal mistrust [52,53]. Similarly, an Australian study discovered that the presence of an eating disorder was related to majored scores on “vulnerable child” mode [27], which portrays people who often feel sad, anxious, dubitative of themselves, and victimized [54]. Moreover, within the literature, mistrust appeared to be largely incriminated in substance abuse disorders, including alcohol dependence and alcohol/opiate abuse [55,56,57,58]. These observations support that mistrust/abuse might be a relevant common feature between eating and substance use disorders, rendering this maladaptive schema a potential transdiagnostic process across various maladaptive behavioral patterns.

Furthermore, besides demonstrating that early maladaptive schemas predict poor eating outcomes, as hypothesized, our findings highlighted the indirect role of difficulties in regulating one’s emotions within the relationships between maladaptive cognitive structures and disordered eating behaviors. Unlike the disconnection and rejection schema domain that was closely tied to dysfunctional eating, the pathways between the remaining maladaptive schema domains and disordered eating attitudes were only significant when governed by a cognitive–affective paradigm. Certainly, both cognitive and emotional factors are pivotal when delineating the psychology of eating. Prior research revealed that, contrasted to women with a healthy body weight, female candidates for bariatric surgery held cognitive misrepresentations and lacked emotional competencies; hence, they were enormously vulnerable to unhealthy eating habits [59]. Another study showed that the disordered eating symptomatology displayed tight linkages with dysfunctional thinking processes and negative emotions; on the other hand, early maladaptive schemas were firmly related to depression and restrained eating patterns [60].

In reality, comparably to early maladaptive schemas that are rigid cognitive dimensions, difficulties in affect regulation are defined by an impaired flexibility in modulating emotional responses, which might engender numerous psychopathologies (e.g., personality disorders, post-traumatic stress disorders, mood and anxiety disorders, interpersonal trauma, etc.) [34,61,62]. Interestingly, existing literature has evidenced that both might correlate with antecedents of abusive childhood experiences and thus are presumed to share intimate endophenotypical features [23,61]. Nonetheless, the standard therapeutic approaches for psychological disorders, such as targeted cognitive interventions, suffer from constraints [63], as promoting cognitive competencies does not strictly imply the endorsement of emotion regulation [64,65,66]. Therefore, Dadomo et al. evidenced the utility of schema therapy—which fosters emotion regulation—as they had explicitly underscored the strong correlation of early maladaptive schemas with dysregulated emotions and problematic emotion regulation strategies that contribute to the persistence of disturbed emotional reactions [63]. Moreover, they innovated a schema therapy model that incorporates relational therapeutic and emotion targeted-experiential techniques [63], addressing the fundamental pillars of functional emotion regulation [34]. Thus, in light of the abovementioned facts and the results of the current study, this innovative schema therapy could be a promising therapeutic approach for eating disorders, alongside personality disorders and other psychiatric conditions.

Lastly, in regard to sociodemographic characteristics, our analysis showed that the higher the household crowding index of the participants, indicating lower socioeconomic-status households, the lower their eating pathology scores. This finding converges to previously reported results in Lebanon [9,67], which is currently enduring a devastating economic crisis. The shortage in incomes is likely to shift disadvantaged Lebanese people’s priorities to procuring their basic nutritional needs, such that they would choose the cheapest food products and focus on surviving this financial crisis rather than dieting and worrying about their weight.

### 4.1. Strengths and Limitations

To the best of our knowledge, our study is the first Lebanese investigation into the relationship between early maladaptive schemas and disordered eating attitudes, and it is a pioneer attempt in research to examine the indirect role of emotion regulation difficulties within this complex interplay. Nonetheless, acknowledging its limitations is of paramount importance, in order to strengthen further similar explorations. As our findings were drawn from a cross-sectional investigation, they are only able to infer pure associations rather than causal relationships. Rigorous longitudinal studies are warranted to delve further into this topic. Additionally, the studied variables were assessed by self-report scales, probably leading to a potential misreport of symptoms, thus occasioning an information bias. The relationships of early maladaptive schemas with eating attitudes might have also been influenced by some residual confounding variables not considered in the current research. For instance, despite the household crowding index being a valid way to assess the socioeconomic status [39], sensitive information about the current income of the participants was not gathered. In addition, since we aimed at elucidating which of the five maladaptive schema domains and the eighteen specific maladaptive schemas would be the most relevant to eating pathology, no variable reduction method has been applied to select the study variables; hence, all subscales were straightly included in the analysis. Finally, it is worth stating that caution is essential when generalizing our findings to the whole Lebanese population, owing to the fact that this study relied on data collected through an online snowball sampling technique. Nonetheless, given the large number of reached participants, we believe in the solidity and reliability of our results.

### 4.2. Clinical Implications

Our results provide support for cognitive models of eating disorders, endorsing that maladaptive schemas’ therapeutic approaches hold the promise of enhancing eating disorder treatment and prevention [68]. Furthermore, the findings sustain the vitalness of interventions that promote emotion regulation skills and encourage the inclusion of targeted trainings in patient-supporting programs. Our study indeed offers a first evidence for the existence of indirect pathways between maladaptive cognitive schemas and eating disorders via defective emotion regulation competencies. Thus, it broadens the medical community’s insights into the underpinning processes behind eating disorder psychopathology and could therefore help make a step forward to the consideration of innovative approaches in the context of schema therapy.

## 5. Conclusions

Our findings gave prominence to a potential intrinsic mechanism through which maladaptive cognitive schemas are linked to disordered eating behaviors, emphasizing the role of emotion dysregulation as a cardinal actor within this model. They sustain the surmise that cognitively and emotionally vulnerable individuals exhibit stronger propensities for inappropriate dietary patterns, as a means to offset their inner weakness. This study provides the groundwork for further research that would work towards reproducing our results in longitudinal samples and strives to extend this mediational model to other components of psychological distress.

## Figures and Tables

**Figure 1 ijerph-19-11620-f001:**
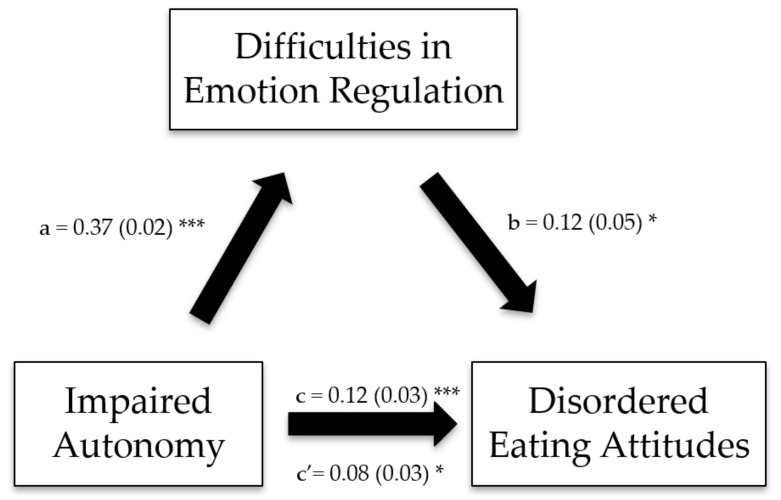
(**a**) Relation between the impaired autonomy schema domain and difficulties in emotion regulation; (**b**) relation between difficulties in emotion regulation and disordered eating attitudes; (**c**) total effect of the impaired autonomy schema domain on disordered eating attitudes; (**c′**) direct effect of the impaired autonomy schema domain on disordered eating attitudes. Numbers are displayed as regression coefficients (standard error). * *p* < 0.05; *** *p* < 0.001.

**Figure 2 ijerph-19-11620-f002:**
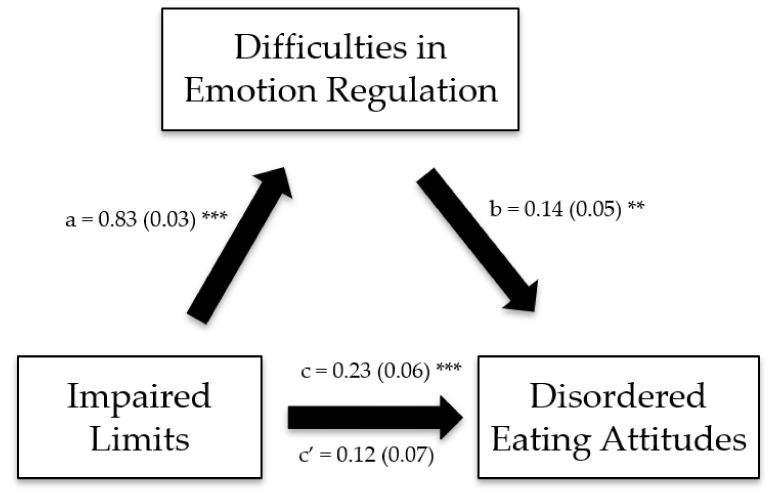
(**a**) Relation between the impaired limits schema domain and difficulties in emotion regulation; (**b**) relation between difficulties in emotion regulation and disordered eating attitudes; (**c**) total effect of the impaired limits schema domain on disordered eating attitudes; (**c′**) direct effect of the impaired limits schema domain on disordered eating attitudes. Numbers are displayed as regression coefficients (standard error). ** *p* < 0.01; *** *p* < 0.001.

**Figure 3 ijerph-19-11620-f003:**
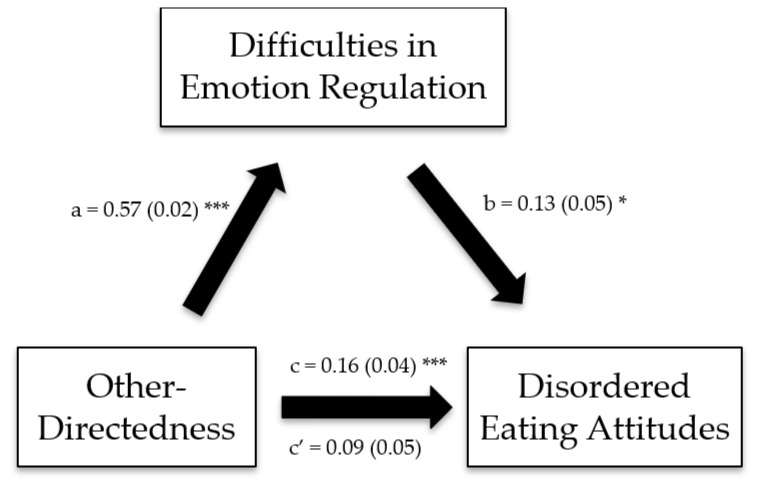
(**a**) Relation between the other-directedness schema domain and difficulties in emotion regulation; (**b**) relation between difficulties in emotion regulation and disordered eating attitudes; (**c**) total effect of the other-directedness schema domain on disordered eating attitudes; (**c′**) direct effect of the other-directedness schema domain on disordered eating attitudes. Numbers are displayed as regression coefficients (standard error). * *p* < 0.05; *** *p* < 0.001.

**Figure 4 ijerph-19-11620-f004:**
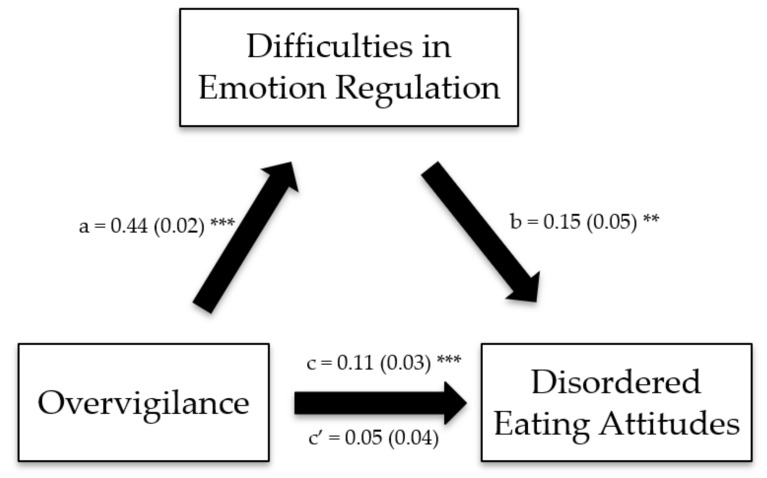
(**a**) Relation between the overvigilance schema domain and difficulties in emotion regulation; (**b**) relation between difficulties in emotion regulation and disordered eating attitudes; (**c**) total effect of the overvigilance schema domain on disordered eating attitudes; (**c**′) direct effect of the overvigilance schema domain on disordered eating attitudes. Numbers are displayed as regression coefficients (standard error). ** *p* < 0.01; *** *p* < 0.001.

**Table 1 ijerph-19-11620-t001:** Sociodemographic characteristics of the participants and scores description (N = 982).

Categorical Variables	N (%)
**Gender**	
Male	281 (28.6%)
Female	701 (71.4%)
**Marital status**	
Single	868 (88.4%)
Married	114 (11.6%)
**Education level**	
Secondary or less	188 (19.1%)
University	794 (80.9%)
**Continuous Variables Mean ± SD**	
Age (in years)	21.97 ± 3.33
Household crowding index	1.14 ± 0.64
Physical activity index	26.23 ± 19.32
Eating Attitudes Test (EAT-26)	22.67 ± 15.12
Emotional Deprivation	15.12 ± 7.37
Abandonment	16.13 ± 7.06
Mistrust	16.97 ± 6.61
Social Isolation	15.26 ± 7.12
Defectiveness	13.27 ± 7.44
Failure	13.41 ± 7.41
Dependence	13.50 ± 7.22
Vulnerability to Harm	14.90 ± 6.96
Enmeshment	15.09 ± 6.70
Subjugation	13.45 ± 7.37
Self-sacrifice	17.67 ± 6.70
Emotional Inhibition	15.15 ± 6.83
Unrelenting Standards	18.20 ± 6.27
Entitlement	17.94 ± 6.49
Insufficient Self-Control	15.63 ± 6.67
Approval Seeking	17.25 ± 6.68
Negativity	16.67 ± 6.84
Punitiveness	17.38 ± 6.20
Disconnection and Rejection Schema Domain	76.75 ± 32.61
Impaired Autonomy/Performance Schema Domain	56.90 ± 26.76
Impaired Limits Schema Domain	33.57 ± 12.31
Other-Directedness Schema Domain	48.38 ± 18.44
Overvigilance/Inhibition Schema Domain	67.41 ± 23.90
Difficulties in Emotion Regulation (DERS-16)	27.67 ± 16.66

**Table 2 ijerph-19-11620-t002:** Reliability analysis of the scores.

Variables	Cronbach’s Alpha
Eating Attitudes Test (EAT-26)	0.96
Emotional Deprivation	0.90
Abandonment	0.88
Mistrust	0.85
Social Isolation	0.89
Defectiveness	0.92
Failure	0.93
Dependence	0.91
Vulnerability to Harm	0.87
Enmeshment	0.83
Subjugation	0.92
Self-sacrifice	0.85
Emotional Inhibition	0.86
Unrelenting Standards	0.81
Entitlement	0.82
Insufficient Self-Control	0.86
Approval Seeking	0.86
Negativity	0.87
Punitiveness	0.81
Disconnection and Rejection Schema Domain	0.97
Impaired Autonomy/Performance Schema Domain	0.97
Impaired Limits Schema Domain	0.90
Other-Directedness Schema Domain	0.94
Overvigilance/Inhibition Schema Domain	0.95
Difficulties in Emotion Regulation (DERS-16)	0.96

**Table 3 ijerph-19-11620-t003:** Correlations between continuous variables and the Eating Attitudes Test score.

Variables	r
Emotional Deprivation	**0.15**
Abandonment	**0.14**
Mistrust	**0.17**
Social Isolation	**0.15**
Defectiveness	**0.16**
Failure	**0.14**
Dependence	**0.14**
Vulnerability to Harm	**0.13**
Enmeshment	**0.15**
Subjugation	**0.15**
Self-sacrifice	**0.11**
Emotional Inhibition	**0.13**
Unrelenting Standards	**0.10**
Entitlement	**0.12**
Insufficient Self-Control	**0.14**
Approval Seeking	**0.12**
Negativity	**0.13**
Punitiveness	**0.10**
Disconnection and Rejection Schema Domain	**0.17**
Impaired Autonomy/Performance Schema Domain	**0.15**
Impaired Limits Schema Domain	**0.14**
Other-Directedness Schema Domain	**0.14**
Overvigilance/Inhibition Schema Domain	**0.13**
Difficulties in Emotion Regulation	**0.15**
Age	0.05
Gender	−0.04
Physical activity index	0.06
Household crowding index	**−0.13**
Marital status	0.05
Education level	−0.02

r = Pearson correlation coefficient; numbers in bold indicate significant *p*-values.

**Table 4 ijerph-19-11620-t004:** Bivariate analysis of categorical variables associated with the Eating Attitudes Test score.

Variables	EAT-26 Score (Mean ± SD)	*p*
**Gender**		0.267
Male	23.87 ± 22.70	
Female	22.19 ± 21.00	
**Marital status**		0.154
Single	22.31 ± 21.46	
Married	25.37 ± 21.73	
**Education level**		0.589
Secondary or less	23.43 ± 21.17	
University	22.49 ± 21.59	

**Table 5 ijerph-19-11620-t005:** Multivariable analyses.

**Model 1: Linear regression taking the EAT-26 score as the dependent variable and** **all maladaptive schemas as independent variables.**				
**Variable**	**Beta**	**β**	** *p* **	**95% CI**
Mistrust	0.55	0.17	**<0.001**	0.35–0.75
Household Crowding Index	−12.26	−0.13	**<0.001**	−18.2–−6.28
Variables entered in the model: all maladaptive schemas, age, marital status, physical activity, household crowding index.				
**Model 2: Linear regression taking the EAT-26 score as the dependent variable and** **all maladaptive schema domains as independent variables.**				
**Variable**	**Beta**	**β**	** *p* **	**95% CI**
Disconnection and Rejection Schema Domain	0.11	0.17	**<0.001**	0.07–0.15
Household Crowding Index	−12.4	−0.13	**<0.001**	−18.2–−6.28
Variables entered in the model: all schema domains, age, marital status, physical activity, household crowding index.				

Beta = unstandardized beta; β = standardized beta; CI = confidence interval; EAT = Eating Attitudes Test. Numbers in bold indicate significant *p*-values.

**Table 6 ijerph-19-11620-t006:** Mediation analysis: Direct and indirect effects of the associations between maladaptive schema domains (independent variables), difficulties in emotion regulation (mediator), and Eating Attitudes Test score (dependent variable).

Independent Variables	Direct Effect			Indirect Effect		
Maladaptive Schema Domains	Effect	SE	P	Effect	SE	95% BCi
Disconnection and Rejection	0.07	0.03	0.006	0.03	0.02	−0.003–0.07
Impaired Autonomy/Performance	0.08	0.03	0.017	0.04	0.02	**0.01–0.08**
Impaired Limits	0.12	0.07	0.095	0.11	0.04	**0.03–0.20**
Other-Directedness	0.09	0.05	0.067	0.07	0.03	**0.02–0.14**
Overvigilance/Inhibition	0.05	0.04	0.180	0.06	0.02	**0.02–0.11**

Numbers in bold indicate significant indirect effects. Direct effect = effect of the maladaptive schema domain on eating attitudes in the absence of the mediator; indirect effect = effect of the maladaptive schema domain on eating attitudes in the presence of the mediator (difficulties in emotion regulation); SE = standard error; BCi = bootstrap confidence interval.

## Data Availability

The authors do not have the right to share data about the respondents as per their institutions policies.

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
