# Peer review of "Maladaptive Cognitive Schemas as Predictors of Disordered Eating: Examining the Indirect Pathway through Emotion Regulation Difficulties"

_ijerph, 2022, doi:10.3390/ijerph191811620_

Round 1

Reviewer 1 Report

In their manuscript entitled "Maladaptive Cognitive Schemas as Predictors of Disordered Eating: Examining the Indirect Pathway through Emotion Regulation Difficulties", the authors tested whether in a general Lebanese cohort emotional dysregulation played the role of mediator between maladaptive schemas and disordered eating. The study was conducted online through the administration of self-report questionnaires. Results confirmed a relationship between several domains of maladaptive schemas and disturbed eating mediated by difficulties in emotional regulation.

The study is interesting and shows that emotional regulation skills are a key modulating factor between cognitive aspects and behavior. The sample seems adequate for the type of study and the methodology appears correct (although all questionnaires but one were not validated in the authors' language, which is a major limitation). Some comments that might improve the manuscript follow.  

Introduction

Lines 62-74: This paragraph could be simplified and shortened.

Line 89: The authors should be more cautious in their remarks. Certainly maladaptive beliefs play an important role in the development of negative thoughts about shape and weight, but these are also generated by other factors related to these psychopathological conditions.

Line: 119: what do the authors mean by “inferior emotional intelligibility”?

Methods

Line 161-165: Could the authors provide some more information on how the translation occurred? Has the value of Cronbach's Alpha obtained from their results been compared to that of the original scales? What is meant by "expert committee"?  

Line 168-169: Could the authors provide some more information about the procedure for administering the questionnaires? Was a service such as Google Forms or Qualtrics used? Was there any incomplete data? How many people in total were sent the questionnaire?

Line 171: Is there any evidence to suggest using the household crowding index as reliable measure of socioeconomic status?

Results

Table 1. Tables clarity can be improved, e.g. “Mean & SD” is on the same row of other variables. I would suggest to report all variables instead of reporting only one in case of dichotomy (e.g. gender).

Table 2: I suppose that here the authors show results correlational results between all these variables and EAT-26. I would make it clear that in the figure caption.

Paragraph 3.2: I understand the authors' rationale in wanting to include all subscales of the questionnaires in their analysis, but they should also justify this choice in light of adjusting for multiple comparisons. Perhaps a method of variable reduction (e.g. Principal Component Analysis) would have been useful to partially obviate this problem. At least, that is to be included as a limitation of the study.

Table 3: Table caption can be improved. Captions should always declare what it’s reported in the table.

Discussion

Line 363: I would replace "anorexic eating disorder patients" with "people with diagnosis of anorexia."

Line 394: Authors should be a little more cautious in their claims. The origin is not always related to childhood abuse trauma.

Author Response

Reviewer 1

In their manuscript entitled "Maladaptive Cognitive Schemas as Predictors of Disordered Eating: Examining the Indirect Pathway through Emotion Regulation Difficulties", the authors tested whether in a general Lebanese cohort emotional dysregulation played the role of mediator between maladaptive schemas and disordered eating. The study was conducted online through the administration of self-report questionnaires. Results confirmed a relationship between several domains of maladaptive schemas and disturbed eating mediated by difficulties in emotional regulation.

The study is interesting and shows that emotional regulation skills are a key modulating factor between cognitive aspects and behavior. The sample seems adequate for the type of study and the methodology appears correct (although all questionnaires but one were not validated in the authors' language, which is a major limitation). Some comments that might improve the manuscript follow.  

Thank you for taking the time to review our manuscript. We appreciate your feedback.

Introduction

Lines 62-74: This paragraph could be simplified and shortened.

We shortened it.

Line 89: The authors should be more cautious in their remarks. Certainly maladaptive beliefs play an important role in the development of negative thoughts about shape and weight, but these are also generated by other factors related to these psychopathological conditions.

The sentence was toned down, “one of” instead of “the core essence”, acknowledging the presence of other factors.

Line: 119: what do the authors mean by “inferior emotional intelligibility”?

Lesser emotional clarity: we replaced intelligibility by clarity.

Methods

Line 161-165: Could the authors provide some more information on how the translation occurred? Has the value of Cronbach's Alpha obtained from their results been compared to that of the original scales? What is meant by "expert committee"?

The Cronbach’s alpha are now reported in a separate table (table 2) and “all scores showed excellent Cronbach’s alpha values”, suggesting their reliability in Arabic comparably to the original scales. As mentioned, the translation was carried out in 2 steps: from English to Arabic, then vice-versa, by two independent translators. An expert committee (constituted of psychologists and psychiatrists) compared the initial English versions to the back-translations to verify intellectual content. We added those information to the manuscript.

Line 168-169: Could the authors provide some more information about the procedure for administering the questionnaires? Was a service such as Google Forms or Qualtrics used? It was done on Google forms. We added this information.

Was there any incomplete data? No since all questions were required. We added this information in the statistical analysis paragraph 2.5.

How many people in total were sent the questionnaire? We cannot know that with Google forms.

Line 171: Is there any evidence to suggest using the household crowding index as reliable measure of socioeconomic status?

Yes, a reference is already inserted in this regard:

  1. Melki IS, Beydoun HA, Khogali M, Tamim H, Yunis KA. Household crowding index: A correlate of socioeconomic status and inter-pregnancy spacing in an urban setting. Journal of Epidemiology and Community Health. 2004;58:476-480. doi:10.1136/jech.2003.012690.

Results

Table 1. Tables clarity can be improved, e.g. “Mean & SD” is on the same row of other variables. I would suggest to report all variables instead of reporting only one in case of dichotomy (e.g. gender).

Corrected as requested.

Table 2: I suppose that here the authors show results correlational results between all these variables and EAT-26. I would make it clear that in the figure caption.

Clarified in the title as follows:

Table 3. Correlations between continuous variables and the eating attitude score.

Paragraph 3.2: I understand the authors' rationale in wanting to include all subscales of the questionnaires in their analysis, but they should also justify this choice in light of adjusting for multiple comparisons. Perhaps a method of variable reduction (e.g. Principal Component Analysis) would have been useful to partially obviate this problem. At least, that is to be included as a limitation of the study.

We included your idea in the limitations paragraph.

Table 3: Table caption can be improved. Captions should always declare what it’s reported in the table.

Added as requested.

Discussion

Line 363: I would replace "anorexic eating disorder patients" with "people with diagnosis of anorexia."

Corrected.

Line 394: Authors should be a little more cautious in their claims. The origin is not always related to childhood abuse trauma.

The sentence was also toned down.

Reviewer 2 Report

This is an interesting study on the relationship between maladaptive cognitive schemes and eating behavior disorders, focusing especially on the role that emotional regulation difficulties may have.

The conclusion of the results obtained was that: A predominance of the "disconnection and rejection" scheme, and of the "distrust" scheme, acquired early, were the ones that best predicted possible eating disorders.

The introductory section offers a very good explanation of the state of the art and the central role of emotional problems in the development of eating disorders. The authors describe the relationship between cognitive schemes and emotions, the correlation between eating pathologies and psychological discomfort or dissatisfaction with the body; as well as the contribution of personality traits or difficulties in affective relationships.

Regarding the Methodology section:

The fact of having used an online questionnaire, disseminated by snowball, is considered appropriate (given the circumstances of the COVID pandemic), but this does not allow certain conclusions to be drawn that are made at the end of the paper.

In the materials sub-section, questionnaires are described, such as: EAT-26, YSQ-SF3, or DERS-16. The authors explain that it haves been translated into the Arabic language and validated for the Lebanese population. We consider it very important to explain the psychometric properties of these questionnaires and, if they are not available, perhaps reliability and validity data obtained from the sample used for this study could be offered.

Regarding the fact of calculating the economic level of the participants through the number of people and number of rooms in the home, it is a valid way, but it should be emphasized that it is "very approximate" to know their status. This fact does not allow certain statements to be made in the discussion/conclusions section, so they should be qualified.

Statistic analysis:As mentioned, if the questionnaires have been used for the first time in the Lebanese population, it would be convenient to offer Cronbach's alpha data that, surely, coincide with the initial psychometric data.Their explanation for why only the Pearson correlations statistic was used, and why they don't use regression (although this statistic is also based on correlations), is not well understood.All this does not allow a causal relationship to be established, but only a correlational one, so the authors should better explain the mediational model they propose.In the Results section:the authors should review table 1, since there is a "jump" of data (women/men).In the Conclusions section.The conclusion they establish between the scheme of mistrust and substance abuse is not understood (357-360). Although they provide explanations, we consider that the leap they take is very large. We believe that they should express it as a "possibility" supported by the literature, but, due to the method and sample of the study, it is not possible to affirm such a conjecture.Finally, the study is very interesting due to its clinical implications, but its great limitations should be underlined in the text, as well as the need to delve much further, with more rigorous methodology, into this subject.

Author Response

Reviewer 2

This is an interesting study on the relationship between maladaptive cognitive schemes and eating behavior disorders, focusing especially on the role that emotional regulation difficulties may have.

The conclusion of the results obtained was that: A predominance of the "disconnection and rejection" scheme, and of the "distrust" scheme, acquired early, were the ones that best predicted possible eating disorders.

The introductory section offers a very good explanation of the state of the art and the central role of emotional problems in the development of eating disorders. The authors describe the relationship between cognitive schemes and emotions, the correlation between eating pathologies and psychological discomfort or dissatisfaction with the body; as well as the contribution of personality traits or difficulties in affective relationships.

Thank you for your comments; you helped us improve our paper.

Regarding the Methodology section:

The fact of having used an online questionnaire, disseminated by snowball, is considered appropriate (given the circumstances of the COVID pandemic), but this does not allow certain conclusions to be drawn that are made at the end of the paper.

We acknowledge our limited capacity of generalizing our statements to the whole Lebanese population (in the limitations paragraph), but as you mentioned, this was our only option during those circumstances.

In the materials sub-section, questionnaires are described, such as: EAT-26, YSQ-SF3, or DERS-16. The authors explain that it haves been translated into the Arabic language and validated for the Lebanese population. We consider it very important to explain the psychometric properties of these questionnaires and, if they are not available, perhaps reliability and validity data obtained from the sample used for this study could be offered.

These were added in table 2 as requested.

Regarding the fact of calculating the economic level of the participants through the number of people and number of rooms in the home, it is a valid way, but it should be emphasized that it is "very approximate" to know their status. This fact does not allow certain statements to be made in the discussion/conclusions section, so they should be qualified.

Yes, as you said, there is a reference suggesting that the household crowding index is a valid measure to estimate the socioeconomic status:

  1. Melki IS, Beydoun HA, Khogali M, Tamim H, Yunis KA. Household crowding index: A correlate of socioeconomic status and inter-pregnancy spacing in an urban setting. Journal of Epidemiology and Community Health. 2004;58:476-480. doi:10.1136/jech.2003.012690.

People with more crowded households may be considered less advantageous, suggesting potentiated struggles in surviving the current economic Lebanese crisis.

Your idea was added as a limitation.

Statistic analysis: As mentioned, if the questionnaires have been used for the first time in the Lebanese population, it would be convenient to offer Cronbach's alpha data that, surely, coincide with the initial psychometric data. Their explanation for why only the Pearson correlations statistic was used, and why they don't use regression (although this statistic is also based on correlations), is not well understood. All this does not allow a causal relationship to be established, but only a correlational one, so the authors should better explain the mediational model they propose.

-We added Cronbach’s alpha values in Table 2 now.

-We removed the partial correlations and replaced them with linear regressions.

-Of course, we cannot establish a causal relationship; this idea is already added in the limitations.

In the Results section: the authors should review table 1, since there is a "jump" of data (women/men).

Yes, thank you. We adjusted the table.

In the Conclusions section. The conclusion they establish between the scheme of mistrust and substance abuse is not understood (357-360). Although they provide explanations, we consider that the leap they take is very large. We believe that they should express it as a "possibility" supported by the literature, but, due to the method and sample of the study, it is not possible to affirm such a conjecture.

We toned the sentence down.

Finally, the study is very interesting due to its clinical implications, but its great limitations should be underlined in the text, as well as the need to delve much further, with more rigorous methodology, into this subject.

We added this idea in the limitations paragraph, following the sentence that expresses our incapacity to infer causality in the current study.

Reviewer 3 Report

Review for

Maladaptive Cognitive Schemas as Predictors of Disordered Eating: Examining the Indirect Pathway through Emotion  Regulation Difficulties

This study aims to the complex relationships between core beliefs, emotional regulation and disordered eating.

Abstract:

  1. The last sentence of the abstract should be the way the results could be used in clinical settings and not an abstract sentence concerning future research.

Introduction:

  1. The Introduction is well written. Although the English itself is correct, there are many places in which the writing is unclear and should be edited by an English speaker, to make it more understandable. Examples: Using the word subjects to describe people or “pieces of research”.
  2. The introductions should end with the study hypotheses.

Methods:

  1. This section is well written.

Results:

  1. It is hard to follow the results as there were no hypotheses in the Introduction.
  2. Sample description belongs to the Methods section.
  3. There should be a separate table in the Results for study indices.
  4. The correlations are very low. I would suggest first assessing which demographic indices have correlations with the study’s indices. Then I would assess the relationships in regression models. I do not agree with the authors explanation for using partial correlations instead of regression.
  5. The authors need to explain their findings in each of the tables and figures. Tables and figures cannot stand alone. There are readers who are less proficient in statistics.

Discussion:

  1. The discussion needs to be edited by an English-speaking editor. It is hard to follow due to the English.
  2. Clinical implications should come at the end

Author Response

Reviewer 3:

This study aims to the complex relationships between core beliefs, emotional regulation and disordered eating.

Thank you for your time, efforts, and comments.

Abstract:

  1. The last sentence of the abstract should be the way the results could be used in clinical settings and not an abstract sentence concerning future research.

We replaced the last sentence in the abstract by the following: “This study broadens the medical community’s insights into the underpinning processes behind eating disorder psychopathology and could therefore help make a step towards the adoption of innovative therapeutic approaches that promote emotion regulation skills in the context of schema therapy.”

Introduction:

The Introduction is well written. Although the English itself is correct, there are many places in which the writing is unclear and should be edited by an English speaker, to make it more understandable.  The introduction was edited.

Examples: Using the word subjects to describe people or “pieces of research”.

“Subjects” was replaced by “people” and “pieces of research” by “studies”.

  1. The introductions should end with the study hypotheses.

Hypotheses were added as follows: “We were working towards identifying the maladaptive schemas/schema domains that would best predict eating behavior disorders; additionally, we hypothesized that emotion regulation problems act as a key mediating factor between such dysfunctional cognitive aspects and eating pathology.”

Methods:

  1. This section is well written.

Thank you J

Results:

  1. It is hard to follow the results as there were no hypotheses in the Introduction.

We added the research hypotheses.

  1. Sample description belongs to the Methods section.

Moved as requested.

  1. There should be a separate table in the Results for study indices.

We removed them from the table and inserted in a separate table as requested.

  1. The correlations are very low. I would suggest first assessing which demographic indices have correlations with the study’s indices. Then I would assess the relationships in regression models. I do not agree with the authors explanation for using partial correlations instead of regression.

We added the bivariate analysis results and those pertaining to the linear regressions.

  1. The authors need to explain their findings in each of the tables and figures. Tables and figures cannot stand alone. There are readers who are less proficient in statistics.

We added information below each table and figure for better clarity.

Discussion:

  1. The discussion needs to be edited by an English-speaking editor. It is hard to follow due to the English.

The discussion was edited for English.

  1. Clinical implications should come at the end

We moved the Clinical Implications paragraph to figure at the end.